# High spatial fidelity among foraging trips of Masked Boobies from Pedro Cays, Jamaica

**Bradley P. Wilkinson** [1]*, **Ann M. Haynes-Sutton**[2], **Llewelyn Meggs**[3], **Patrick G. R. Jodice**[4]

**1** Department of Forestry and Environmental Conservation, South Carolina Cooperative Fish and Wildlife Research Unit, Clemson University, Clemson, South Carolina, United States of America, **2** Marshall's Pen, Mandeville, Jamaica, **3** Yardie Environmental Conservationists Limited, Kingston, Jamaica, **4** Department of Forestry and Environmental Conservation, U.S. Geological Survey South Carolina Cooperative Fish and Wildlife Research Unit, Clemson University, Clemson, South Carolina, United States of America

* bpwilki@g.clemson.edu

**Data Availability Statement:** The data underlying the results presented in the study are available from ScienceBase at https://doi.org/10.5066/P9AK95EG.

## Abstract

In marine environments, tropical and subtropical habitats are considered to be inherently less productive than more temperate systems. As such, foraging site fidelity among vertebrate predators occupying low-latitude marine systems is generally low as a response to an increased unpredictability of resources. We investigated the foraging movements of Masked Boobies breeding on Middle Cay, Jamaica using GPS loggers to examine if the presence of a nearby bathymetric feature influenced foraging site fidelity in a tropical system, the Caribbean Sea. According to the movements of tracked individuals, this population of boobies shows a high degree of spatial fidelity in foraging site selection, concentrated on the northern edge of Pedro Bank. We suggest this feature as an important location for marine conservation in the region and demonstrate its utility to foraging boobies via habitat modeling using a maximum entropy approach of relevant habitat variables. Finally, we place this study into the global context of Masked Booby foraging by examining the published literature of relevant tracking studies for population-level similarity in foraging metrics. According to hierarchical clustering of foraging effort, Masked Boobies demonstrate a density-dependent response to foraging effort regardless of colony origin or oceanic basin consistent with the principles of Ashmole's Halo.

## Introduction

Tropical marine systems are considered to be unproductive relative to high- and mid- latitude systems, with the former characterized by a heterogeneous and patchy distribution of limited resources [1]. Unlike temperate systems, where productivity driven by larger mesoscale oceanographic features such as boundary fronts or coastal upwelling is temporally-focused yet predictable [2–4], tropical and subtropical resources are scarce, seasonally-diffuse, and associated with dynamic and smaller sub-mesoscale features [5, 6]. For predators in tropical marine environments, resources may therefore be less predictable spatially and temporally in relation to temperate systems.

**Funding:** Funding and support for data collection, field logistics, and analysis was provided by The National Fish and Wildlife Foundation, The Nature Conservancy, and the South Carolina Cooperative Fish and Wildlife Research Unit. Yardie Environmental Conservationists Limited currently provided support in the form of salary for author LM but did not have any additional role in the study design, data collection and analysis, decision to publish, or preparation of the manuscript. The specific roles of these authors are articulated in the 'author contributions' section.

**Competing interests:** The authors have declared that no competing interests exist. The affiliation of LM with Yardie Environmental Conservationists Limited does not alter our adherence to PLOS ONE policies on sharing data and materials.

As a means to offset lower predictability of prey resources spatially and temporally, tropical seabirds have developed physiological and behavioral strategies to improve flight performance [7] and optimize foraging efficiency [8]. Generally, these correspond to increases in aerial proficiency at the expense of other methods of locomotion (e.g. swimming or diving capabilities) [9]. Many species also utilize facilitated foraging, wherein individuals forage over subsurface predators that aggregate prey near the surface, creating inherently ephemeral and dynamic feeding opportunities [8, 10, 11]. Given a lack of spatial and temporal predictability of prey resources in tropical systems, foraging behavior of seabirds may be predicted to be more dispersive, multi-directional, and characterized by spatially expansive searches compared to that of temperate or high-latitude seabirds. These issues may be compounded for central-place foragers, which are limited in their movements by a fixed spatial location and dependent upon transient patches of quality habitat capable of producing a net energetic gain that are inherently unpredictable in nature [12, 13].

In addition to environmental features, density dependent factors can also impact how an organism forages for resources. For colonial seabirds, resources proximal to the location of breeding can be depleted, with individuals being forced to invest in foraging trips of greater distance of duration [14–16]. This effect, known as Ashmole's halo [17], should increasingly impact individuals in a population as the number of congeners increases [18]. In tropical environments, where resources are already patchy and limited, density dependent factors may serve to limit population sizes [19]. In theory, foraging effort should therefore be positively associated with population size in central-place foragers such as seabirds.

We used GPS loggers to study the daily movements of Masked Boobies (*Sula dactylatra*) breeding on Middle Cay, Jamaica, a small islet located at Pedro Bank in the central Caribbean Sea. Occupying a pantropical distribution, this species has been the subject of a relatively large number of movement-based studies globally (e.g. Table 1). However, our understanding of seabird ecology and marine habitat use is limited in the Caribbean region, at both species and community levels [20, 21,22]. Our objectives were to 1) identify marine habitat used by seabirds in this under-studied region of the Caribbean; 2) quantify the degree of repeatability in foraging behavior exhibited by Masked Boobies from Middle Cay; and 3) place the foraging effort of this population into global context by examining how intraspecific density-dependent factors may impact this species on a pantropical level. We hypothesized that the presence of Pedro Bank, a regionally-important bathymetric feature, would significantly influence the foraging movements of Masked Boobies from Middle Cay by increasing productivity relative to surrounding marine habitats via oceanographic processes [6, 23, 24]. We also predicted that global foraging effort across the range of the species would positively increase with colony size based on density-dependent factors limiting resource availability close to breeding sites [25].

**Table 1. Average foraging metrics of Masked Boobies acquired via literature search used in a hierarchical clustering analysis.**

| Colony | Year | No. of individuals | Trip duration (hrs) | Total distance (km) | Maximum distance (km) | Study |
|---|---|---|---|---|---|---|
| Clipperton Is. | 2005 | 120000 | 8.9 | 103.1 | 107.0 | Weimerskirch et al. [23] |
| Islas Muertos | 2013–2014 | 5000–6000 | 10.1 | 192.3 | 71.6 | Poli et al. [6] |
| Ascension Is. | 2011, 2013–2014 | 4600 | 11.4 | 199.0 | 78.0 | Oppel et al. [25] |
| Anguilla | 2014 | 680 | 3.0 | 61.1 | 23.9 | Soanes et al. [21] |
| Phillip Is. | 2010 | 600 | 6.5 | 160.7 | 74.4 | Sommerfeld et al. [26] |
| St. Helena Is. | 2014 | 500 | 3.4 | 118.0 | 41.0 | Oppel et al. [25] |
| Tromelin Is. | 2005–2006 | 400–500 | 5.3 | 135.5 | 45.8 | Kappes et al. [27] |
| Middle Cay | 2012 | 100–150 | 4.6 | 60.8 | 18.6 | *current study* |
| Palmyra Atoll | 2008 | 20–100 | 2.8 | 89.4 | 29.4 | Young et al. [28] |

## Methods

### Data collection

We collected spatial data at the Masked Booby colony on Middle Cay (17˚01', 77 ˚47') in the Pedro Bank group, Jamaica. We visited the colony (approx. 100–150 individuals) from 26 June—2 July, and again from 15–22 October, 2012, as part of a training workshop designed to enhance capacity for seabird science and management in the Caribbean. IGotU Global Positioning System (GPS) tags were attached to Masked Boobies rearing chicks at Middle Cay using standard tagging protocols [6]. Chicks ranged in developmental stage from downy to downy with some flight feather emergence. Briefly, birds were captured by hand at the nest site and carried < 200 m to a staging area for processing. Each bird was assessed for general condition, weighed, and measured (culmen, wing chord, tarsus). Any unbanded birds were banded with USGS Bird Banding Laboratory bands and unsealed white Darvic leg bands to assist with resighting and recovery. GPS tags (IGotU, Mobile Action Technology, Taiwan) measured 30mm x 45mm x 15mm and weighed 21g (mass of individual boobies $\geq$ 1100g). Tags were encased in latex condoms for waterproofing and attached to the underside of the tail at the base of the tail using Tesa Tape. We programmed devices to record a location every 3 min except when birds were flying >20 km/h during which time locations would be recorded every 30 sec. All birds were returned to their nest sites for release within 15 minutes of capture. Deployments were generally < 5 days in duration depending on recovery effort and ability to resight tagged individuals. Field research was conducted with permission from the Clemson University Animal Care and Use Committee (2012–009) and the U.S. Geological Survey Bird Banding Laboratory (22408).

### Data processing

GPS devices were removed from the adult upon recapture and data downloaded using software provided by the manufacturer. We used the package *adehabitatLT* [29] in the R statistical framework for data processing. Booby tracks were filtered for erroneous locations using a speed threshold of 95 km·h$^{-1}$ [27]. All recorded locations were kept for further analysis. Trip segmentation was determined using a 1 km threshold from the colony in the package *trakR* [30], and points were rediscretized at a 180 sec interval. Only locations from complete trips were used for measures of trip characteristics and site fidelity. All locations, regardless of trip completion, were used in habitat analyses.

### Trip characteristics and repeatability

Trip duration, total distance travelled, and maximum distance from the colony were calculated for each complete trip using the *trip* package in R [31]. As a measure of trip repeatability at the population level, we used the Fidelity Index (*FI*) applied by Shaffer et al. [32] based on a model from Hazen et al. [33]. Briefly, the distance and angle of displacement from the colony to the furthest point were calculated for each trip. Due to the small sample size of complete trips (n = 11), some individuals contributed more than a single trip (range = 1–3). We then calculated the population means for distance and angle of displacement (using circular statistics for displacement angles) across all trips. The normalized difference between individual trip distance ($dist_i$) and mean trip distance ($dist_m$) was then summed with the normalized difference between individual angular displacement ($angle_i$) and mean angular displacement ($angle_m$) using the formula

$$FI = 2 \times [(dist_i - dist_m) \div dist_i] + [(angle_i - angle_m) \div 90]$$

for each individual trip. The formula results in a score ranging from 0–4 for each trip, with values closer to 0 indicating higher similarity for an individual trip to the population mean (high fidelity) and values closer to 4 indicating lower similarity for an individual trip to the population mean (low fidelity). We then averaged *FI* from each trip to acquire a population mean of trip fidelity across individuals.

## Species distribution modeling

The at-sea distribution of Masked Boobies from Middle Cay was explored using a maximum entropy approach implemented in the open-source software Maxent v. 3. 4.1. (Phillips et al. [34]; http://biodiversityinformatics.amnh.org/open_source/maxent/). Maxent generates species distribution models that estimate the density of spatially-discrete environmental covariates conditioned on presence-only animal data (e.g. telemetry data), and is capable of producing useful results at comparatively small sample sizes compared to alternative methods [35]. Environmental variables of interest included bathymetry, bathymetric slope, mean annual sea surface salinity (SSS), annual variance in SSS, mean annual sea surface temperature (SST), and annual variance in SST. All variables were downloaded at a 30 arcsecond resolution from the MARSPEC data platform (http://marspec.weebly.com/modern-data.html, *accessed 4/19/2019*) and encompassed the entire study area [36]. Environmental variables were chosen based on probable relevancy to booby distribution as well as spatial scale [6]. Due to the comparatively local movements of Masked Boobies from this population, we considered only those remotely-sensed variables able to discriminate local oceanographic features and excluded other variables collected at relatively coarse scales (e.g. chlorophyll-*a*). In addition, we chose to model environmental variables on an annual temporal scale due to the large amount of time separating bouts of data collection (several months) and to identify persistent habitat features likely to be present throughout the reproductive period. Default parameters in the Maxent interface were used for analysis (500 iterations). Model performance was evaluated by fitting species occurrence data into training (80%) and test (20%) datasets using the random test percentage setting. Models were calibrated with training data and evaluated using test data via area under the receiver operating characteristics curve (AUC). Contributions of each environmental variable to the final model were evaluated with a jackknife procedure.

## Global clustering of foraging effort

We used a hierarchical clustering approach to examine potential similarities between movement characteristics of Masked Boobies rearing chicks at Middle Cay with colonies located in the Atlantic, Pacific, and Indian Ocean basins. Trip parameters were acquired from the literature (Table 1) after a thorough search using both Google Scholar and ScienceDirect using combinations of keywords 'Masked Booby; tracking; transmitter; GPS; and foraging'. Only studies using GPS devices were used; those using PTTs were discarded to improve trip comparability. *A priori* movement characteristics of interest were trip duration, maximum distance from the colony per trip, and total trip distance. These metrics were chosen based on widespread availability and as a relative proxy of foraging effort. Average values for each parameter were obtained from each colony, and were restricted to the chick-rearing stage of breeding, as boobies may undertake trips of different lengths and durations depending on breeding stage [24]. Estimates of Masked Booby population size were also recorded from each study. The presence of other seabird species at each colony was not considered as reliable estimates of population sizes were frequently unreported.

Trip characteristics were then scaled at the global level and used to create a distance matrix between colonies using Pearson correlation coefficients. A k-means clustering analysis was

then performed on the resultant dissimilarity values. Optimal number of clusters was decided using gap statistics (bootstrapped to 100 iterations), and validated using Dunn's Index. Per the maximization of Dunn's Index, clustering was performed using $k = 6$ clusters with a random start of n = 25 chosen sets. Finally, Euclidean relationships between colonies were examined using a hierarchical dendrogram produced via criterion from [37]. Visual assessment of the resultant dendrogram was used to relate overall trip metrics (a proxy for foraging effort) to population size between and among study colonies.

## Results

### Distribution

Data were successfully obtained from 8 Masked Boobies, resulting in 2700 unique locations comprising 21 trips. Of these, 11 trips were considered complete (no data gaps and clearly defined departures and arrivals). Boobies were distributed almost exclusively to the north of Middle Cay, with the majority of movements occurring over Pedro Bank (Fig 1). Boobies dispersed from and returned to the colony daily, with movements contained to diurnal hours. Trips appeared directed to the northern edge of Pedro Bank, where lateral movements to the east and west along the break were common, preceding a relatively rapid return to the colony.

### Fidelity index

Visual inspection of complete trips indicated a spatial similarity in terminus points of foraging boobies. Bearings from the colony to the point of maximal distance were similar across trips, with 72% of trips ending between 345˚—45˚ (i.e., NW to NE of the colony, Fig 2). Maximum

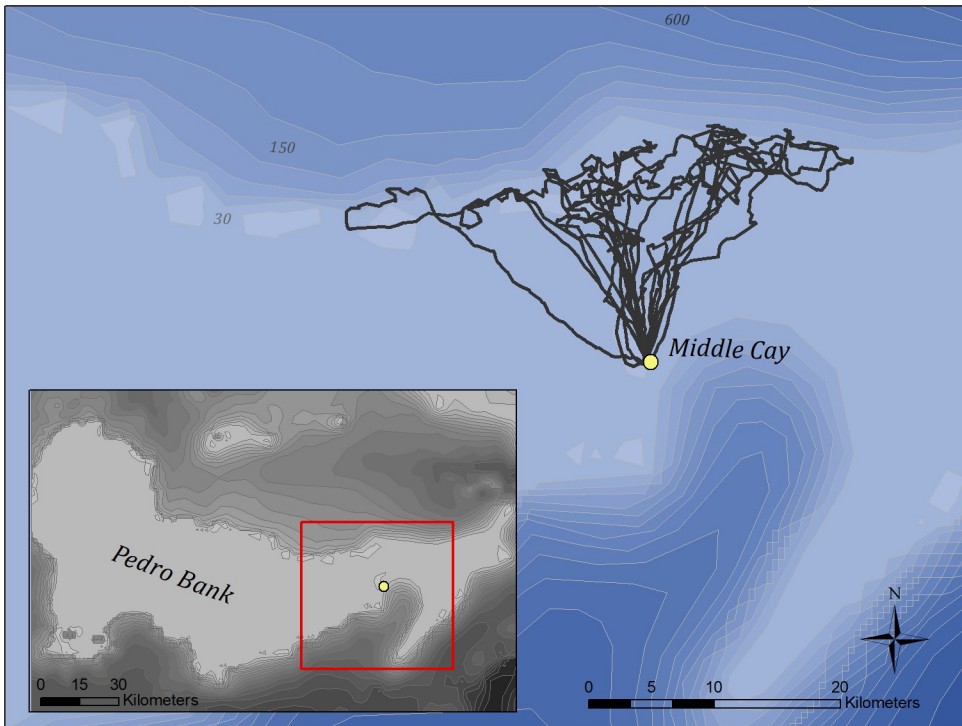

**Fig 1. Complete foraging trips of GPS-equipped Masked Boobies from Middle Cay, Jamaica (yellow marker).** Gray lines represent bathymetric gradients, with darker colors indicating an increase in depth. Numbers represent approximate isobaths.

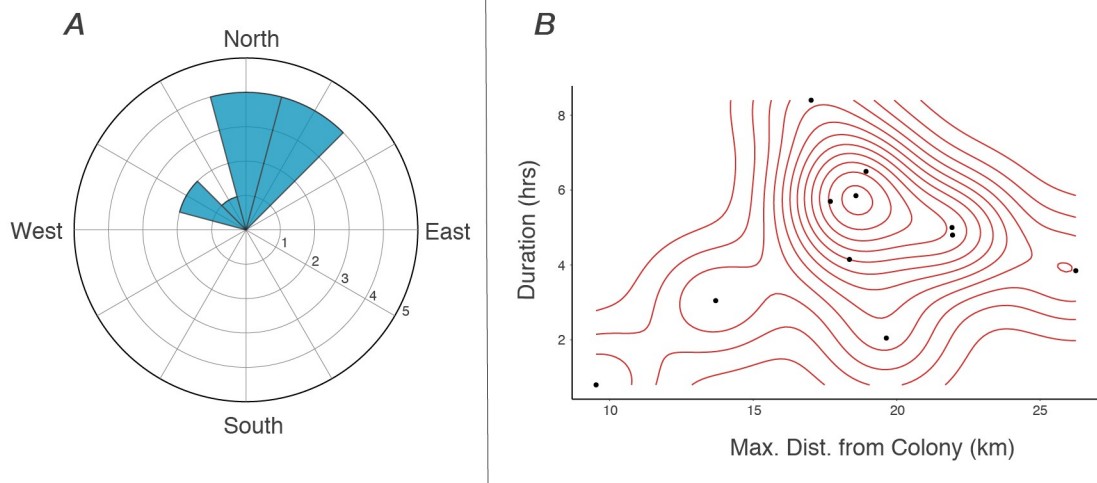

**Fig 2.** A) Rose plot depicting the relative angle from Middle Cay, Jamaica, to the point of maximal distance for each complete trip taken by foraging Masked Boobies tracked with GPS loggers. Bins are segmented into 30° intervals; numbers represent number of trips taken within each bin. B) Distribution of foraging durations and maximum distances from the colony for each complete trip recorded by Masked Boobies from Middle Cay, Jamaica. Red lines represent a relative landscape of foraging effort as determined via kernel density.

distances and durations were also similar, with the majority of points ≤ 20 km from the colony and most trips ≤ 6 hrs duration, respectively (Table 1). Fidelity Index scores per trip ranged from 0.11–1.69, with a population mean of 0.72 ± 0.50, suggesting a relatively high degree of fidelity per individual trip relative to the overall population.

## Species distribution modeling

Pedro Bank was highlighted as highly suitable habitat for foraging Masked Boobies, concentrated on the eastern half surrounding Middle Cay (Fig 3). Model performance (AUC = 0.96 ± 0.005) indicated good ability of the model to predict booby distribution. Jackknife procedures indicated mean annual sea surface salinity (39.8%), annual variance in sea surface salinity (30.0%), and bathymetry (23.9%) to be the highest environmental contributors to the final model. All other variables possessed relative contributions < 5.0%. Permutation importance was 10.0 for mean annual SSS, 49.3 for annual variance in SSS, and 17.8 for bathymetry. Probability of occurrence showed a negative relationship with mean annual SSS and bathymetry, and a curvilinear relationship with annual variance in SSS (S1 Fig).

## Global clustering of foraging effort

K-means clustering of population-level foraging effort in Masked Boobies (n = 9 colonies) indicated a relationship between foraging metrics (duration, total distance, and maximum distance) and population size. According to gap statistics, $k = 6$ clusters were chosen as optimal (Dunn's Index = 0.83). Although clustered using indices of foraging effort, dendrogram results aligned colonies of similar population sizes together (Fig 4). Apparent relationships between foraging effort and colony size were evident in both nodes (within-cluster) and branches (between-cluster) of the dendrogram.

## Discussion

The distribution of foraging locations for Masked Boobies breeding on Middle Cay, Jamaica, during this study was highly influenced by the presence of Pedro Bank, a unique and

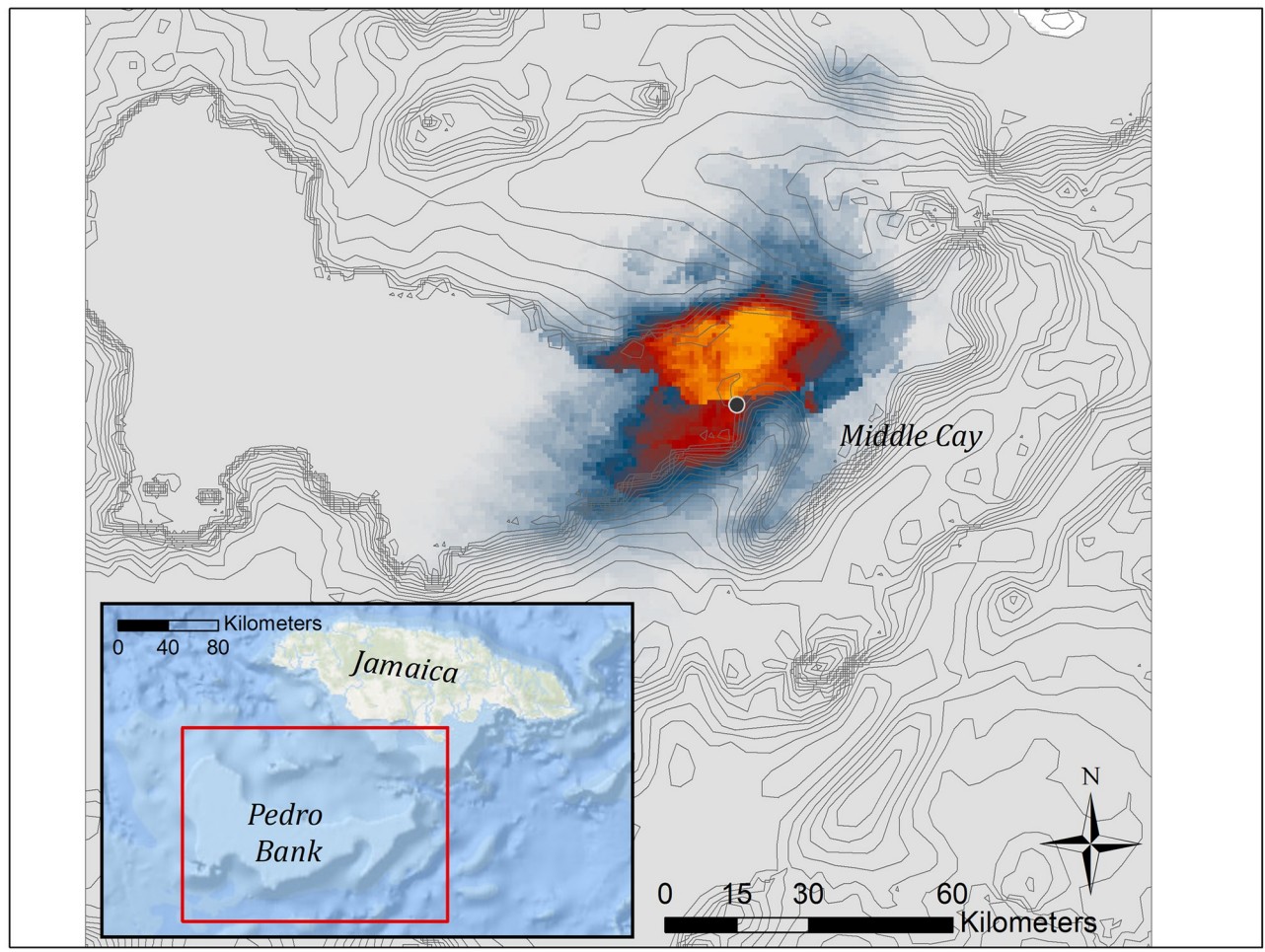

**Fig 3. Output of Maxent habitat modeling for Masked Boobies from Middle Cay based on a suite of oceanographic figures.** Areas of high suitability are represented by increasingly yellow colors. Lower use areas are represented by increasingly blue colors.

prominent bathymetric feature in the region. Compared to other populations of this species, boobies from Middle Cay display a relatively high degree of spatial fidelity among foraging trips at the colony level, focused on the northern edge of Pedro Bank. This area was also highlighted as important marine habitat when modeled using relevant oceanographic indices, underscoring the value of this feature to the surrounding ecosystem. When examined across the pantropical distribution of the species, foraging trips undertaken from this colony were comparatively truncated in both duration and distances travelled, matching with the globally small population size of boobies on Middle Cay.

Located within the Greater Antilles marine ecoregion [38], Pedro Bank is an ecologically and economically valuable underwater feature approximately 80 km from the southern edge of the Jamaican mainland [39]. Supporting the most productive fisheries in the country, primarily targeting queen conch (*Lobatus gigas*), spiny lobster (*Panularis sp.*), and finfish, Pedro Bank and associated small cays host a diverse and abundant marine community as well as a seasonally-variable yet significant human population of artisanal fishers [40, 41]. With the largest human settlement on Middle Cay, interactions between nesting boobies and local fishers are likely common and widespread (A. Haynes-Sutton pers. obs.). However, characterizing these interactions, especially at-sea, remains unresolved and difficult to assess.

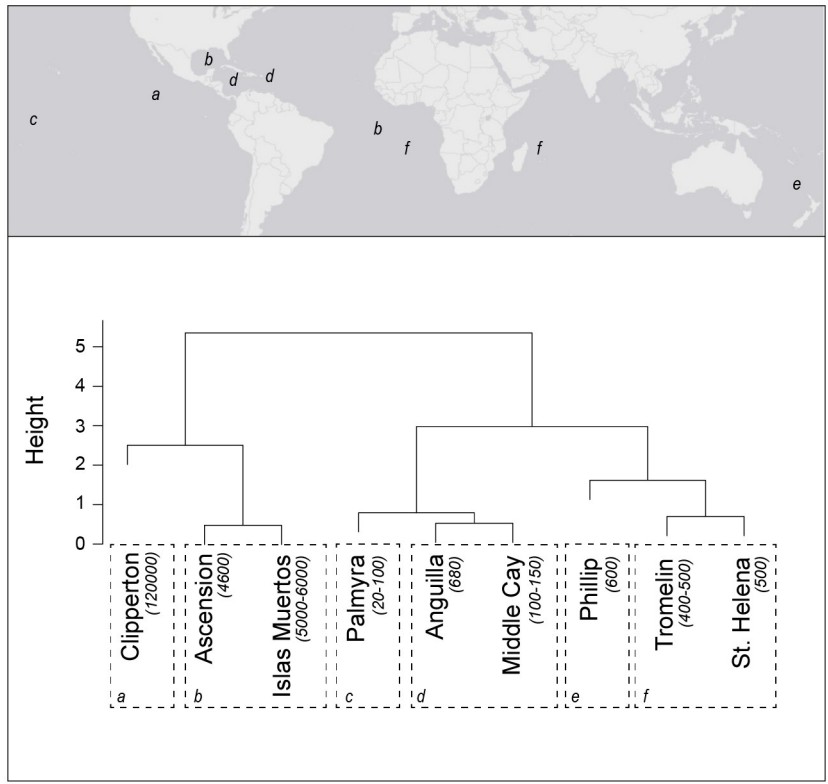

**Fig 4. Hierarchical clustering analysis of global Masked Booby foraging effort derived from GPS tracking studies.**
Population estimates of breeding individuals are shown in italics, with letters identifying significant clusters and
approximate locations. Height indicates the Euclidean distance between clusters.

Despite the regional importance of Pedro Bank biologically, to our knowledge this is the
first study explicitly linking this area to the movements of a top marine predator. The promi-
nence of Pedro Bank as critical habitat, particularly to Masked Boobies, is underpinned by the
high fidelity observed in this study. We suggest that boobies from Middle Cay display repeat-
ability in foraging trips at the population level due to a relatively predictable increase in pro-
ductivity formed by the underlying bathymetry of the area. Previous examinations of the
hydrodynamic environment surrounding Jamaica have found sea surface velocities within the
study area to be highly variable in both intensity and direction, with the formation of many
mesoscale currents, eddies, and jets that are ephemeral in both space and time [41]. A notable
exception is Pedro Bank, over which sea surface velocities are significantly depressed. Impor-
tantly, this area of low water movement is relatively stable throughout the annual cycle [41].

We posit that this comparatively predictable boundary between low, stable water velocities
over Pedro Bank and the markedly complex currents surrounding it acts as an area of
enhanced productivity for foraging boobies, functionally increasing trip fidelity in this colony.
Marine habitats characterized by sharp gradients in water velocity often aggregate prey, espe-
cially in tropical environments, and may serve as local 'hotspots' for marine predators (e.g.
Chambault et al. [42]). Indeed, previous work by Poli et al. [6] highlighted the importance
of oceanographic gradients in sea surface height and velocity to foraging Masked Boobies in
the southern Gulf of Mexico over Campeche Bank, results that appear to be consistent with
ours. Critically, however, boobies from Middle Cay can exploit a static feature inducing

oceanographic gradients, instead of adopting a dispersive and widely-searching strategy common in other populations for encountering favorable foraging conditions.

Habitat modeling also delineated Pedro Bank as important marine habitat, focusing on the eastern half of the feature surrounding Middle Cay (Fig 3). Interestingly, portions of the southern edge of the bank also were featured in model outputs, although this area was largely unused by tracked boobies in this study. This region appears to possess many of the same oceanographic features of the more heavily utilized northern edge in terms of the tested environmental variables, but could vary in other aspects of oceanography or productivity not apparent in our analysis. Direction of travel may also be influenced by dominant wind patterns, providing an energetic benefit to concentrate foraging on the northern edge of Pedro Bank. It should also be noted that relatively coarse predictor variables were used temporally as a tradeoff for increased spatial resolution when modeling. As such, we urge caution in assigning specific oceanographic variables as causal to booby distribution in this analysis. Instead, we aim only to provide a generalized distribution of foraging boobies while at sea built on the tracks of focal individuals. We also emphasize that our model was built with the input of relatively few individuals, and that preferred foraging habitat can vary based on a number of intrinsic and extrinsic factors (e.g. Sommerfeld et al. [23]). However, we consider the results useful although constructed with a relatively small sample size [35].

When placed into global context, boobies from Middle Cay undertake foraging trips of comparatively short duration and distance. The most closely matched colony displaying equivalent measures of foraging effort was on nearby Anguilla, which clustered together with Middle Cay on a hierarchical dendrogram (Fig 4). These colonies have population estimates on the same order of magnitude (Anguilla supports approximately 680 individuals annually) and are geographically proximate (~ 1500 km apart). Palmyra Atoll, in the central Pacific Ocean and the smallest in terms of population, comprised its own cluster and was placed on the same branch when examined in the context of foraging effort. The next most closely related branch is comprised of three colonies with very similar population sizes (Phillip Island, Tromelin Island, and St. Helena; 400–600 individuals annually) yet distributed in differing oceanic basins. Finally, the last branch contains the largest colonies; two of similar size (Ascension Island and Islas Muertas; 4600–6000 individuals annually) and an outlying cluster occupied by Clipperton Island, which is orders of magnitude larger than any other colony (120000 individuals annually).

Despite withholding population information in our k-means clustering analysis, when organized by foraging effort colonies also clearly segregate by number of individuals present. We suggest this as evidence for Ashmole's halo in Masked Boobies on a global scale. Although Masked Boobies are distributed pantropically, occupying vastly different ocean basins, foraging effort as measured by trip distance and duration scales positively with local population size regardless of location and concomitant marine ecoregion. While evidence for Ashmole's halo has been observed in this species from neighboring colonies occupying similar systems [25], findings from this study underscore the global context under which population regulation may occur by means of density dependence. It also highlights the value of collecting and publishing relevant tracking data from across the entire range of a species, especially in understudied regions such as the Caribbean. It must be noted, however, that boobies may vary trip distances and durations depending on breeding stage (e.g. early versus late chick-rearing), and although all foraging data used in this analysis originated from individuals rearing chicks, exact stage could not be controlled [24].

We suggest that the foraging patterns of Masked Boobies from Middle Cay are subject to interactive factors of both the nearby environment and local density dependent processes found on a global scale. Disentangling the relative contribution of each to individual boobies

may be critical to understanding the continuance of this colony in the face of growing anthropogenic pressure. Establishing programs aimed at monitoring seabird populations in the Pedro Bank region may serve to provide this information.

## Supporting information

**S1 Fig. Estimated relationships between oceanographic variables identified via Maxent and Masked Booby habitat suitability.** Note missing decimals in plots of mean sea surface salinity and variance of sea surface salinity.
(DOCX)

## Acknowledgments

The Jamaica Defence Force provided transportation to and from Pedro Cays. The tagging was conducted as part of a National Fish and Wildlife Foundation capacity building workshop with BirdsCaribbean and assistance in the field with tagging and handling was provided by the workshop trainees; Damany Calder, Martha-Ines Garcia-Escobar, Jaedon Lawe, Adriana Vallarino Moncada, Maureen Milbourn, Carine Precheur, Fernando Simal, Hugh Small, Camillo Trench, and Susan Zaluski. We are grateful to Esther Figueroa, Vagabond Media/Juniroa Productions, Inc., who diligently filmed all aspects of the research and training effort. The South Carolina Cooperative Fish and Wildlife Research Unit is jointly supported by the U.S. Geological Survey, South Carolina DNR, and Clemson University. Any use of trade, firm, or product names is for descriptive purposes only and does not imply endorsement by the U.S. Government.

## Author Contributions

**Conceptualization:** Ann M. Haynes-Sutton, Patrick G. R. Jodice.

**Data curation:** Patrick G. R. Jodice.

**Formal analysis:** Bradley P. Wilkinson.

**Investigation:** Patrick G. R. Jodice.

**Methodology:** Bradley P. Wilkinson, Patrick G. R. Jodice.

**Project administration:** Llewelyn Meggs, Patrick G. R. Jodice.

**Supervision:** Ann M. Haynes-Sutton, Patrick G. R. Jodice.

**Writing – original draft:** Bradley P. Wilkinson.

**Writing – review & editing:** Bradley P. Wilkinson, Patrick G. R. Jodice.

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
