## [Decision Letter · Decision Letter 0]

24 Feb 2020

PONE-D-19-34930

High spatial fidelity among foraging trips of Masked Boobies from Pedro Bank, Jamaica

PLOS ONE

Dear Mr. Wilkinson,

Thank you for submitting your manuscript to PLOS ONE. After careful consideration, we feel that it has merit but does not fully meet PLOS ONE’s publication criteria as it currently stands. Therefore, we invite you to submit a revised version of the manuscript that addresses the points raised during the review process.

The manuscript is well written, and most of the comments from the reviewers and myself can be easily addressed.

We would appreciate receiving your revised manuscript by Apr 03 2020 11:59PM. To enhance the reproducibility of your results, we recommend that if applicable you deposit your laboratory protocols in protocols.io, where a protocol can be assigned its own identifier (DOI) such that it can be cited independently in the future. For instructions see: http://journals.plos.org/plosone/s/submission-guidelines#loc-laboratory-protocols

We look forward to receiving your revised manuscript.

Kind regards,

William David Halliday, Ph.D.

Academic Editor

PLOS ONE

Additional Editor Comments (if provided):

This study is well written and easy to follow. Both reviewers felt that only minor revisions were required before it could be acceptable for publication. I agree with this assessment.

My only major comments are related to the results. First, using the subheading "Maxent" or "Maxent output" is not useful, unless the reader already knows what this is. Call it habitat modeling or something similar. In the results, please expand on the results of the Maxent model. Provide effect sizes, test statistics, and appropriate means/ranges of values for the different important habitat variables. For example, you state that "a positive relationship with bathymetry" exists, but this is vague at best. This could mean that boobies are foraging in 1 m of water, but clearly this isn't the case. This is where providing ranges of values for the preferred bathymetry would be extremely useful. Same comment applies to SSS. Also, please redefine what SSS is in the results for the sake of the readers.

My other comment is related to the "Hierarchical clustering" results. Again, please use a more descriptive subheading. Second, as with the previous section, provide much more detail in the results. If there is a density-dependent relationship, then what is that density-dependent relationship. I assume, based on the hypothesis presented earlier in the manuscript, that the relationship between colony size and foraging distance was positive, but provide slope estimates at least.

My final comment: in the last paragraph of the discussion, you state that density dependence is a global process. This is simply incorrect. Yes, density dependence can be found globally, but it is inherently driven by local factors (i.e. competition for food within the colony, in this case). Please reword, or fully justify this statement.

Journal Requirements:

"Funding and support for this research was provided by The National Fish and Wildlife Foundation, The Nature Conservancy, and the South Carolina Cooperative Fish and Wildlife Research Unit."

4. Your ethics statement must appear in the Methods section of your manuscript. If your ethics statement is written in any section besides the Methods, please move it to the Methods section and delete it from any other section. Please also ensure that your ethics statement is included in your manuscript, as the ethics section of your online submission will not be published alongside your manuscript.

5. Please include your tables as part of your main manuscript and remove the individual files. Please note that supplementary tables (should remain/ be uploaded) as separate "supporting information" files

Reviewers' comments:

Reviewer's Responses to Questions

**Comments to the Author**

1. Is the manuscript technically sound, and do the data support the conclusions?

Reviewer #1: Partly

Reviewer #2: Yes

2. Has the statistical analysis been performed appropriately and rigorously? 

Reviewer #1: Yes

Reviewer #2: Yes

3. Have the authors made all data underlying the findings in their manuscript fully available?

Reviewer #1: Yes

Reviewer #2: Yes

4. Is the manuscript presented in an intelligible fashion and written in standard English?

Reviewer #1: Yes

Reviewer #2: Yes

5. Review Comments to the Author

Reviewer #1: L 77: better use the number of individuals 100-150 here, as used in Table 1 and Fig. 4

L 85: How much do they birds weigh? Please add this information

L 88-89: How long did this take? Please add this information

L 236-240: you also have a small sample size (n = 8 birds) and only 11 complete trips. Moreover a short sample period. It is known that trip characteristics can vary between seasons (environmental variability), sexes and breeding stage in seabirds, incl. Masked boobies. Please discuss how this may affect your study.

L258-259: Please explain this statement: "regardless of location and surrounding environmental conditions". You only analyzed environmental variables at Middle Cay. Do the other studies analysed how the environment affects trip characteristics?

Table 1: Add "No of individuals" in column 3 Population estimate

Figure 4 Explain the y-axis Height - what does it mean?

Reviewer #2: Globally the manuscript is well written which allows a fluid and comprehensive reading. All sections have enough detail to be fully understandable and replicable.

At the end of the introduction, along with the main objectives of the work, authors should state concrete study hypothesis followed by predicted results emanated from the literature. Those can come from publications on the same study species or related taxa, on the same or similar study system.

Minor comment:

L58 - Replace “…necessitating foraging trips of greater distance or duration.” by “…with individuals being forced to invest in foraging trips of greater distance or duration.”

6. PLOS authors have the option to publish the peer review history of their article (what does this mean?). If published, this will include your full peer review and any attached files.

Reviewer #1: No

Reviewer #2: No

---

## [Author Response · Author response to Decision Letter 0]

26 Mar 2020

March 26th, 2020

Dr. William David Halliday, Academic Editor

PLoS ONE

Dear Dr. Halliday,

It is my pleasure to submit a revised version of our manuscript, PONE-D-19-34930, entitled “High spatial fidelity among foraging trips of Masked Boobies from Pedro Bank, Jamaica”. We have carefully considered each comment and made edits accordingly, which are detailed below. These thoughtful and constructive comments were fully appreciated by all authors, and we hope our edits have contributed well to the overall quality of the manuscript. We look forward to working with you and the reviewers to advance this manuscript to publication. 

Additionally, please alter our funding statement to read, “Funding and support for this research was provided by The National Fish and Wildlife Foundation, The Nature Conservancy, and the South Carolina Cooperative Fish and Wildlife Research Unit.” Thank you.

Sincerely,

Bradley Wilkinson (on behalf of all authors)

Responses to reviewer comments:

Academic Editor

This study is well written and easy to follow. Both reviewers felt that only minor revisions were required before it could be acceptable for publication. I agree with this assessment.  (i) My only major comments are related to the results. First, using the subheading "Maxent" or "Maxent output" is not useful, unless the reader already knows what this is. Call it habitat modeling or something similar. In the results, please expand on the results of the Maxent model. Provide effect sizes, test statistics, and appropriate means/ranges of values for the different important habitat variables. For example, you state that "a positive relationship with bathymetry" exists, but this is vague at best. This could mean that boobies are foraging in 1 m of water, but clearly this isn't the case. This is where providing ranges of values for the preferred bathymetry would be extremely useful. Same comment applies to SSS. Also, please redefine what SSS is in the results for the sake of the readers.

Relevant subheadings have been changed to ‘Species distribution modeling’ to improve clarity. We have provided all relevant test statistics provided by Maxent, per the software instructions developed by the provider. In addition, we have formatted results to closely match those of other recently-published manuscripts that use Maxent (e.g. Geary, B., Leberg, P. L., Purcell, K. M., Walter, S. T., & Karubian, J. (2020). Breeding Brown pelicans improve foraging performance as energetic needs Rise. Scientific Reports, 10(1), 1-9.). However, we have included a Supplementary Figure depicting the relationships graphically between the top-performing models and booby habitat use as determined by Maxent. We also emphasize and urge caution in assigning specific oceanographic variables as causal to booby distribution in this analysis due to our sample size and variables used, as stated in L266.   (ii) My other comment is related to the "Hierarchical clustering" results. Again, please use a more descriptive subheading. Second, as with the previous section, provide much more detail in the results. If there is a density-dependent relationship, then what is that density-dependent relationship. I assume, based on the hypothesis presented earlier in the manuscript, that the relationship between colony size and foraging distance was positive, but provide slope estimates at least.

Subheadings have been changed from ‘Hierarchical clustering’ to Global clustering of foraging effort’ to improve clarity. Hierarchical clustering analysis does not provide slope estimates of relationships, and we have provided all relevant test statistics in the manuscript. For clarity, we did not examine a direct relationship between foraging effort and colony size in a model-based approach (e.g. using linear regression). Instead, we clustered colonies by only foraging effort, which grouped colonies that displayed similar effort together. We then examined how these clusters related to colony size, and found that clusters (built on only information of foraging effort) were apparently structured by colony size as well.   (iii) My final comment: in the last paragraph of the discussion, you state that density dependence is a global process. This is simply incorrect. Yes, density dependence can be found globally, but it is inherently driven by local factors (i.e. competition for food within the colony, in this case). Please reword, or fully justify this statement.

Yes, thank you for this comment. We acknowledge the error in scale in our description of density dependence processes. We have reworded to say, “ …Boobies from Middle Cay are subject to interactive factors of both the nearby environment and local density dependent processes found on a global scale.”

Reviewer #1

 L 77: better use the number of individuals 100-150 here, as used in Table 1 and Fig. 4.

We have changed colony size estimates from “pairs” to “individuals” for continuity in the manuscript.

L 85: How much do the birds weigh? Please add this information.

Due to various equipment failures, we were unable to obtain mass estimates for many individuals although all were weighed. However, we are confident that all individuals weighed greater than 1100g, which we have added to the manuscript.

L 88-89: How long did this take? Please add this information.

All deployment procedures were completed in < 15 minutes. This has been added to the manuscript.

L 236-240: you also have a small sample size (n = 8 birds) and only 11 complete trips. Moreover a short sample period. It is known that trip characteristics can vary between seasons (environmental variability), sexes and breeding stage in seabirds, incl. Masked boobies. Please discuss how this may affect your study.

Yes, we agree that sample size is a limitation of this study. We have added the following text to the manuscript. “We also emphasize that our model was built with the input of relatively few individuals, and that preferred foraging habitat can vary based on a number of intrinsic and extrinsic factors (e.g. Sommerfeld et al. 2015). However, we consider the results useful although constructed with a relatively small sample size (Hernandez et al. 2006). 

L258-259: Please explain this statement: "regardless of location and surrounding environmental conditions". You only analyzed environmental variables at Middle Cay. Do the other studies analysed how the environment affects trip characteristics?

We acknowledge the lack of clarity in this statement, and have changed the sentence to read, “…regardless of location and concomitant marine ecoregion.” We suggest that marine ecoregions have inherently different oceanographic properties, but we did not analyze environmental conditions at any other location and are simply using global location as a proxy. 

Table 1: Add "No of individuals" in column 3 Population estimate.

We have changed the column header from “Population estimate” to “No. of individuals” for clarity.

Figure 4: Explain the y-axis Height - what does it mean?

Height refers to the Euclidean distance between clusters, as defined in the hierarchical clustering analysis. We have added this information to the figure caption.

Reviewer #2

Globally the manuscript is well written which allows a fluid and comprehensive reading. All sections have enough detail to be fully understandable and replicable.

(i) At the end of the introduction, along with the main objectives of the work, authors should state concrete study hypothesis followed by predicted results emanated from the literature. Those can come from publications on the same study species or related taxa, on the same or similar study system.

We have included two statements outlining hypotheses for the study, as well as additional citations supporting hypotheses relating to the predicted importance of Pedro Bank as well as the influence of density-dependent processes upon the species. 

L58 - Replace “…necessitating foraging trips of greater distance or duration.” by “…with individuals being forced to invest in foraging trips of greater distance or duration.”

The suggested change has been made.

---

## [Editor Report · Decision Letter 1]

30 Mar 2020

High spatial fidelity among foraging trips of Masked Boobies from Pedro Cays, Jamaica

PONE-D-19-34930R1

Dear Dr. Wilkinson,

We are pleased to inform you that your manuscript has been judged scientifically suitable for publication and will be formally accepted for publication once it complies with all outstanding technical requirements.

With kind regards,

William David Halliday, Ph.D.

Academic Editor

PLOS ONE
---

## [Editor Report · Acceptance letter]

1 Apr 2020

PONE-D-19-34930R1 

High spatial fidelity among foraging trips of Masked Boobies from Pedro Cays, Jamaica 

Dear Dr. Wilkinson:

I am pleased to inform you that your manuscript has been deemed suitable for publication in PLOS ONE. Congratulations! Your manuscript is now with our production department. 

With kind regards,

on behalf of

Dr. William David Halliday 

Academic Editor

PLOS ONE